# Antimicrobial Activity of Extracts of Two Native Fruits of Chile: Arrayan (*Luma apiculata*) and Peumo (*Cryptocarya alba*)

**DOI:** 10.3390/antibiotics9080444

**Published:** 2020-07-25

**Authors:** Jitka Viktorová, Rohitesh Kumar, Kateřina Řehořová, Lan Hoang, Tomas Ruml, Carlos R. Figueroa, Monika Valdenegro, Lida Fuentes

**Affiliations:** 1Department of Biochemistry and Microbiology, University of Chemistry and Technology Prague, 166 28 Prague, Czech Republic; Jitka.Prokesova@vscht.cz (J.V.); Rohitesh.Kumar@vscht.cz (R.K.); Katerina.Rehorova@vscht.cz (K.Ř.); Lan.Hoang@vscht.cz (L.H.); tomas.ruml@vscht.cz (T.R.); 2Institute of Biological Sciences, Campus Talca, Universidad de Talca, Talca 3465548, Chile; cfigueroa@utalca.cl; 3Agronomy School, Faculty of Agronomic and Food Sciences, Pontificia Universidad Católica de Valparaíso, Quillota 2260000, Chile; monika.valdenegro@pucv.cl; 4Regional Center for Studies in Healthy Food (CREAS), CONICYT-Regional GORE Valparaíso Project R17A10001, Avenida Universidad, Valparaíso 2340000, Chile

**Keywords:** traditional medicine, drug-resistant bacteria, high-resolution HPLC/MS, biofilm disruption, quorum sensing

## Abstract

Arrayan and peumo fruits are commonly used in the traditional medicine of Chile. In this study, the concentration of the extracts halving the bacterial viability and biofilms formation and disruption of the drug-sensitive and drug-resistant strains of *Staphylococcus aureus* and *Pseudomonas aeruginosa* was determined. The chemical composition of extracts was analyzed by high-resolution liquid chromatography coupled with mass spectrometry (U-HPLC/MS). The arrayan extract (Inhibitory concentration IC_50_ 0.35 ± 0.01 mg/mL) was more effective than peumo extract (IC_50_ 0.53 ± 0.02 mg/mL) in the inhibition of *S. aureus* planktonic cells. Similarly, the arrayan extract was more effective in inhibiting the adhesion (*S. aureus* IC_50_ 0.23 ± 0.02 mg/mL, *P. aeruginosa* IC_50_ 0.29 ± 0.02 mg/mL) than peumo extracts (*S. aureus* IC_50_ 0.47 ± 0.03 mg/mL, *P. aeruginosa* IC_50_ 0.35 ± 0.01 mg/mL). Both extracts inhibited quorum sensing in a concentration-dependent manner, and the most significant was the autoinducer-2 type communication inhibition by arrayan extract. Both extracts also disrupted preformed biofilm of *P. aeruginosa* (arrayan IC_50_ 0.56 ± 0.04 mg/mL, peumo IC_50_ 0.59 ± 0.04 mg/mL). However, neither arrayan nor peumo extracts disrupted *S. aureus* mature biofilm. U-HPLC/MS showed that both fruit extracts mainly possessed quercetin compounds; the peumo fruit extract also contained phenolic acids and phenylpropanoids. Our results suggested that both extracts could be used as natural antimicrobials for some skin and nosocomial infections.

## 1. Introduction

As part of the diets, many fruits of native species have been used as a traditional medicine in the South American region [1,2]. The Chilean trees, arrayan [*Luma apiculata* (DC.) Burret] and peumo [*Cryptocarya alba* (Molina) Looser], are ancestral species used in food and medicinal preparations [3,4]; however, the antimicrobial properties of both species are have not yet been described in detail.

Arrayan (*Luma apiculata*) is a Myrtaceae [5] endemic plant from the Valparaiso to Aysen regions of Chile (latitude 33° to 45° S). It grows in the temperate forests of Chile and Argentina. Its fruit is an edible black or purple berry (Figure 1A), with an intense flavor, aromatic, and a pleasant sweet taste. *Luma* species (namely *L. apiculata* and *L. chequen*) have been previously described as a rich source of phenolic compounds with high antioxidant activity [4]. Extracts of ripe fruits of arrayan showed the presence of flavonol and anthocyanins, with high oxygen radical absorption capacity and concentration-dependent vascular protection under high glucose conditions [6]. Moreover, aqueous extracts demonstrated inhibition of the collagen-induced aggregation of platelets in human blood [7], indicating the potency to treat wounds and inflammations. Further, the addition of dry leaves into the lettuce substratum resulted in better control of *Meloidogyne hapla* population, nematode known as a vegetable pathogen, which produces tiny galls [8]. Arrayan has also shown activity against Herpes simplex virus type 2 [9].

Peumo belongs to the Lauraceae family and is spread from the Maule to Araucania regions of Chile (latitude 35° to 38° S). The ripe fruit is a red to smooth pink berry (Figure 1B), crowned with the remains of the stamens and calyx lobes. It has a large seed similar to a nut. Its leaves and bark are used in traditional medicine to treat liver diseases and rheumatism [10]. The essential oil prepared by the distillation of peumo showed activity against *Nosema ceranae*, the originator of nosemosis in honeybee [11]; against insect *Sitophilus zeamais* Motschulsky, known as a maize weevil, causing significant loses during cereals’ storage [12]; and against *Musca domestica* L. [13]. Moreover, the leaves extract demonstrated a protective effect against ethyl methane sulphonate-induced mutagenesis in *Drosophila melanogaster* [14]. The edible peumo skin has a large number of polyphenolic compounds [15], related to high antioxidant capacity [15,16]. Besides the other biologically active compounds, cryptofolione and its dihydro-derivative were isolated from the fruits of *C. alba* [17]. In the trypanocidal assay, cryptofolione at 250 µg/mL reduced the number of trypomastigotes by 77%, while its dihydro-derivative was inactive. However, cryptofolione at tenfold lower concentration (25 µg/mL) reduced the viability of macrophages, demonstrating a cytotoxic effect instead of selectivity towards *Trypanosoma cruzi* [17].

The antimicrobial activity of fruit extracts, including the native fruits of Chile, has usually been related to total phenolic content and compounds such as chlorogenic acid, quercetin, ellagic acid, and quercetin-3-galactoside [18,19,20,21]. The hydroxyl groups of polyphenolic compounds play a significant role in inhibiting the key enzymatic activities of bacteria and inducing cell toxicity [22].

This study aimed to characterize the antimicrobial activity and the phytochemical profile of fruit-extracts and seek potential uses of arrayan and peumo species in the medicinal treatment.

## 2. Results

### 2.1. Inhibition of Multidrug-Resistant Bacterial Strains

As drug-resistant strains, we used clinical isolates obtained from the Collection of the Laboratory of Medical Microbiology (Czech Laboratory, lnc., NEM) resistant to several drugs. Overall, arrayan extract was effective in inhibiting both the drug-resistant and drug-sensitive strains of *S. aureus*, with almost the same IC_50_ values (Table 1). However, the extract showed activity against neither the drug-sensitive nor the drug-resistant strains of *P. aeruginosa* up to the concentration of 1 mg/mL. In lower concentrations, the beneficial (growth-promoting) effect of the extract was observed. Peumo extract was slightly less effective and did not show any specificity towards the *S. aureus* strains. However, unlike arrayan, the peumo extract exhibited weak activity against the drug-resistant strain of *P. aeruginosa* with the IC_50_ value of 0.778 ± 0.004 mg/mL.

As shown in Table 1, both extracts showed reasonable direct toxicity against drug-resistant strains (higher against the Gram-positive *S. aureus*); therefore, their ability to modulate the drug-resistant phenotype was also tested. Both extracts were applied in a dose equal to IC_25_ concentration obtained for drug-resistant *S. aureus* (240 µg/mL for peumo and 270 µg/mL for arrayan) on both drug-resistant strains *S. aureus* and *P. aeruginosa*). After that, IC_50_ of chloramphenicol in the presence of the extracts was determined and compared with IC_50_ of the antibiotic alone. However, the addition of neither arrayan nor peumo extract decreased the IC_50_ of chloramphenicol (Table 2). Thus, even that both extracts inhibited the drug-resistant strains, their mode of action was instead direct cytotoxicity than selective modulation of drug resistance.

### 2.2. Inhibition of Biofilm Formation

To test the inhibition of biofilm formation, the ability of extracts to inhibit cell adhesion of drug-sensitive strains was determined. Overall, compared with the peumo extract, the arrayan extract was slightly more active against both *S. aureus* and *P. aeruginosa* strains. The arrayan extract was more effective towards *S. aureus* strain (Table 3).

### 2.3. Disruption of Mature Biofilm

Both arrayan and peumo extracts did not disrupt mature biofilm of *S. aureus* (Table 4), contrary to the results observed for both extracts on the mature biofilm of *P. aeruginosa*.

### 2.4. Inhibition of Quorum Sensing

The inhibition of bacterial adhesion is usually connected with the inhibition of bacterial extracellular communication. To verify the anti-adhesion results, inhibition of quorum sensing was tested using two mutant sensor strains of *Vibrio campbellii*, which responds either only to (i) autoinducer type-1 (AI-1) molecules (BAA1118) or (ii) autoinducer type-2 (AI-2) molecules (BAA1119). As can be seen in Table 5, the concentrations halving the viability differed significantly from those halving the communication. The decrease in luminescence was thus caused by inhibition of communication rather than inhibition of cell growth. The peumo extract was more effective in inhibiting AI-1 based communication, which is more typical for Gram-negative bacteria. In contrast, the AI-2 type communication was inhibited rather by the arrayan extract. As the AI-2 type communication is based on boron compounds and is used by both Gram-positive and Gram-negative bacteria, the inhibition caused by arrayan extract is more promising.

### 2.5. High-Resolution Liquid Chromatography Coupled with Mass Spectrometry (U-HPLC/MS) of Arrayan and Peumo Extracts

U-HPLC/MS analysis showed that arrayan samples are rich in polyphenolics such as catechins, quercetins, and myricetins (Figure 2, Table 6). Peumo samples are rich in polyphenolics such as procyanidins, chlorogenic acid and its analogous, catechins, and quercetins. Moreover, this extract shows the presence of (+)−lariciresinol, a phenylpropanoid compound (Figure 3, Table 7). In the extracts of both fruits tested, there were some molecular ions and molecular formula that did not reveal any hits from the SciFinder database in the search using the molecular mass and the sample names. This indicates that these compounds have not yet been reported to occur in these plant species. Therefore, these are labelled as unidentified in the tables.

## 3. Discussion

Both arrayan [*Luma apiculata* (DC.) Burret] and peumo [*Cryptocarya alba* (Molina) Looser] plants belong to the trees of Mediterranean ecosystems in Central Chile [3,4]. However, even though both plants are used as food or in traditional medicine, the lack of knowledge about their biological activities and composition still limits their potential applications.

The present study shows the antimicrobial activity of arrayan and peumo extracts, potentially associated with their chemical profiles. The inhibitory effect of arrayan fruits extract against both drug-resistant and drug-sensitive strains of *S. aureus* was overall higher than the peumo fruit extracts (Table 1, Figure 4). Both extracts showed no inhibitory activity against the drug-sensitive strain of *P. aeruginosa*; however, only the peumo extract showed an effect against the drug-resistant strain of *P. aeruginosa*. Similar results were also reported by Araya-Contreras et al. [23], who reported significant antimicrobial activity of arrayan leaves extract against Gram-positive *S. aureus*, but no activity against Gram-negative strains. On the basis of our results, neither peumo nor arrayan extracts can modulate the resistant phenotype of bacteria to chloramphenicol; therefore, their mode of action should be rather direct cytotoxicity than inhibition of mechanisms leading to multidrug-resistance. The antimicrobial activity has been reported for other species belonging to *Myrtaceae* family of Chile, that is, murta (*Ugni molinae* Turcz.), as its seed extract showed high antibacterial activity against both Gram-positive (*Staphylococcus aureus*, *Streptococcus pyogenes*, and *Bacillus cereus*) and Gram-negative (*Salmonella typhi*, *Escherichia coli*, and *Pseudomonas aeruginosa*) bacterial strains [21]. Owing to the antimicrobial activity against planktonic cells, we also tested the potential of both extracts to disrupt the mature biofilm. Both extracts disrupted the mature biofilm of *P. aeruginosa* in a concentration-dependent manner; however, none of the tested extracts disrupted the mature biofilm of Gram-positive *S. aureus* (Table 4).

Both extracts showed promising activity in inhibition of planktonic and biofilm cells; therefore, their ability to inhibit extracellular communication was also investigated. Firstly, the inhibition of cell adhesion was tested using sensor mutant strains of *Vibrio campbellii,* and subsequently, the results were confirmed by inhibition of cell adhesion mediated by the cell-to-cell communication (Table 5). The two assays confirmed that the better extract for inhibition of bacterial communication is arrayan extract when applied on bacteria using the AI-2 type of communication. On the basis of our knowledge, this is the first paper reporting on the ability of arrayan fruit to inhibit bacterial extracellular communication altering, potentially through downregulation of the expression of genes related to virulence, motility, sporulation, and biofilm formation [25].

In our study, the U-HPLC/MS analysis showed a significant presence of flavonols, principally quercetin, and its derivate in the extracts of arrayan and peumo fruits (Table 6 and Table 7). In arrayan extract, our previous study by High Performance Liquid Chromatography-Diode Array Detector method (HPLC-DAD) showed higher concentration (between 1- and 11-fold differences) of quercetin-3-rutinoside compared with anthocyanins [6]. In the same study, we observed the presence of arabinoside and galactoside glycosides coupled to petunidin, peonidin, and malvidin structures. However, we could not identify anthocyanin compounds in the present work. These results could be caused by the previously published fact [10] that secondary metabolites in the aerial biomass strongly variate in medicinal species. The main compounds identified in the peumo extract were chlorogenic acid and its analogs, catechins, and quercetins. Similarly, Simirgiotis et al. [15] reported flavanol derivatives, flavonol aglycones, phenolic acids, and some of their derivatives, flavonoid *O*-glycosides, without anthocyanins reported for the peumo extract. In agreement with our results, Timmermann et al. [26], also reported flavonoids (isorhamnetin, kaempferol, quercetin, and their glycosides) and chlorogenic acid present in leaves extract of peumo. In contrast to the previous report of Castro-Saavedra et al. [27], no alkaloid was detected by U-HPLC/MS. Both fruit extracts showed the presence of quercetins and their derivatives. Quercetin extracted from plants has been reported to inhibit the growth of a broad spectrum of microorganisms, including filamentous fungi and bacteria, with particular effects on *E. coli* and *S. aureus* strains [28,29,30]. It has been suggested that the significant level of phenolics, proanthocyanidins, and tannins, quantified by colorimetric determinations and HPLC–MS/MS, should contribute to the potential antimicrobial activity of the murta seeds extract [21]. Studies have reported that the effect of the polyphenol is related to its ability to interfere with bacteria biofilm (e.g., *Enterococcus faecalis*) formation at high concentrations [31].

It has also been reported that bioactive fractions rich in tannins, flavonoids, sterols, and glycosides reduced the ampicillin minimum inhibitory concentration against the pathogen [30]. Interestingly, both fruit extracts suppressed the growth of methicillin-resistant *S. aureus*. These results may be owing to the presence of quercetins and their derivatives in both fruit extracts. The quercetin-3-rutinoside has been reported to interact with penicillin-binding protein 2a (PBP2a, a cell-wall synthesizing protein), which causes resistivity in methicillin-resistant *S. aureus* (MRSA) against β-lactam antibiotics [31]. This compound can inhibit PBP2a and thus has the potential to be used in treating MRSA infections [32,33,34].

Finally, all these results suggest different mechanisms of extracts of arrayan and peumo fruits to inhibit bacterial growth and communication that should be investigated in more detail.

## 4. Materials and Methods 

### 4.1. Plant Material

Branches of peumo and arrayan trees, containing leaves and ripe fruits (Figure 1A,B), were collected from Viña del Mar (33°02′18.58″ S; 71°29′48.04″ W; 77 m a.s.l., Valparaiso) and Antuco (37°22′35.01″ S; 71°29′21.20″ W; 850 m a.s.l., Bio-Bio), respectively. The leaves and fruits were manually separated and washed with a sodium hypochlorite solution (100 mg·L^−1^) (J.T. Baker™, Mexico).

### 4.2. Samples Drying and Extracts Preparation

Fruit samples were freeze-dried, finely ground by hand, and pulverized using a mortar. Three grams of each sample was mixed with 50 mL of 100% methanol (J.T. Baker^™^, Mexico) for the extraction [6,35]. The mixture was shaken (Slow-Speed Orbital Shaker, Cole-Palmer, USA) at 100 rpm and room temperature for 24 h. Then, mixtures were passed by cheesecloth to get rid of big particulates and then through a Whatman No. 1 filter (Whatman International Ltda., Maidstone, UK). Filtered extracts were dried under reduced pressure at 40 °C using a rotary evaporator (Büchi, Labortechnik AG, Flawil, Switzerland). The extraction yield was calculated as a percentage of dry extract by 3 g of the initial sample. Crude extracts were reconstituted in methanol to give a concentration of 500 mg/mL, filter-sterilized through a 0.45 µm syringe filter (Millipore, Billerica, MA, USA) and stored at −20 °C.

### 4.3. Inhibition of Multidrug-Resistant Bacterial Strains

The inhibitory activity of both extracts was determined against Gram-positive and Gram-negative bacterial strains. Commercial antibiotics oxacillin and gentamicin (Sigma-Aldrich, USA) were used as a control. Microorganisms were obtained from the Czech Collection of Microorganisms (CCM, Masaryk University, Czech Republic) and the Collection of Laboratory of Medical Microbiology (Czech Laboratory, lnc.): *Staphylococcus aureus* (control/sensitive strain, ATCC 25923), *S. aureus* (multidrug-resistant strain, NEM 449), *Pseudomonas aeruginosa* (control/sensitive strain, ATCC 27853), and *P. aeruginosa* (multidrug-resistant strain, NEM 986). *Staphylococcus aureus* NEM 449 was resistant to gentamicin, clindamycin, erythromycin, chloramphenicol, vancomycin, ciprofloxacin, methicillin, penicillin G, cefotaxime, and tetracycline. *Pseudomonas aeruginosa* NEM 986 was resistant to gentamycin, ciprofloxacin, tetracycline, chloramphenicol, penicillin G, and erythromycin. The resistant or sensitive phenotype was determined by measuring of minimal inhibition concentration (MIC) of each antibiotic by broth dilution method and comparing the MIC value with the clinical breakpoints defined by EUCAST (European Committee on Antimicrobial Susceptibility Testing).

The antibacterial activity of extracts was evaluated by the standard broth-dilution method using 96-well plates and Mueller–Hinton (MH) broth. The overnight microbial culture was diluted to the turbidity equal to 0.5 McFarland. The binary dilutions of the extracts with cell suspension provided the range of tested concentrations from 2 to 1000 µg/mL. The control was microbial suspension without the extract. The plates were incubated for 24 h at 37 °C, and then absorbance (590 nm) was recorded using the SpectraMax i3x Multi-Mode Detection Platform (Molecular Devices, USA). Each inhibitory activity was measured in four replicates. The sensitization of drug-resistant bacteria was realized according to [36].

### 4.4. Inhibition of Biofilm Formation

The effect of extracts on biofilm formation was tested on *S. aureus* (ATCC, 25923) and *P. aeruginosa* (CCM, 3955). The assay was done in 96-well plates. The overnight culture of the tested organism was diluted with brain heart infusion (BHI) broth to obtain the turbidity equal to 0.5 McFarland and split into the plate by the addition of 100 µL to each well. The test samples were added to the wells in concentration range of 0.01–5 mg/mL. The plate was incubated for 24 h at 37 °C. The viability of adherent cells was evaluated immediately by the resazurin assay as follows. The medium was removed, the wells were washed three times by phosphate-buffered saline (PBS), and 100 µL of resazurin in PBS (0.03 mg/L) was added. The viability was evaluated by measuring fluorescence (560/590 nm, ex./em.) using the SpectraMax i3x Multi-Mode Detection Platform (Molecular Devices, USA). Each experiment was done in eight repetitions.

### 4.5. Disruption of Mature Biofilm

We tested the ability of extracts to disrupt mature biofilms formed by monocultures of either *S. aureus* (ATCC, 25923) or *P. aeruginosa* (CCM, 3955). The assay was done in 96-well plates. The overnight cultures of tested microorganisms were diluted with BHI broth to obtain the turbidity equal to 0.5 McFarland and split to the plates into 100 µL aliquots per well. After 24 h of incubation at 37 °C, the medium was removed, and the wells were washed three times by PBS (pH 7.4). Then, fresh BHI broth was added with extracts. After 24 h of incubation, the medium was removed, the wells were washed three times by PBS, and 100 µL of resazurin in PBS (0.03 mg/L) was added. The viability was evaluated by measuring fluorescence (560/590 nm, ex./em.) using the SpectraMax i3x Multi-Mode Detection Platform (Molecular Devices, USA). Each experiment was done in 16 repetitions.

### 4.6. Inhibition of Quorum Sensing

For the evaluation of quorum sensing inhibition activity, two commercial (ATCC) strains of *Vibrio*
*campbellii* were used—BAA1118 and BAA1119. Both strains were cultivated and all experiments were performed in autoinducer bioassay (AB-A) medium composed of NaCl (17.5 g/L), MgSO_4_ (12.3 g/L), casamino acids (2 g/L), 10 mM potassium phosphate (pH 7.0), 1 mM l-arginine, and glycerol (10 mL/L). The overnight culture was diluted with AB-A medium to the density of 0.2 McFarland. This culture was diluted 5000 × with AB-A medium before testing. First, the viability of the *Vibrio campbellii* was determined to set up the experiment at non-toxic concentrations of both extracts. The twofold dilution of extracts was applied on *Vibrio campbellii* in 96-well plate, and after 24 h, the cell viability was determined by resazurin assay. The extracts were applied in the non-toxic concentration and further binary diluted with the cell suspension. After that, the luminescence was recorded for 16 h with a measurement step of 20 min using a microplate reader (SpectraMax i3 Multi-Mode Detection Platform, Molecular Devices, UK) set up at 30 °C; integration time of 10,000 ms; shaking for 60 s prior each measurement. After the measurements, the data were collected, and the sum of relative luminescence units (RLU) (area under the kinetic curve) was calculated and used for the determination of IC_50_. Each experiment was done in four repetitions.

### 4.7. Data Processing and Statistical Analysis

The experiments were done with the appropriate number (n) of repetitions, which are shown for each method. The relative activity was evaluated as a percentage according to the formula: 100 × (slope of sample fluorescence−the average slope of PC)/(average slope of NC−the average slope of PC). IC_50_ values were determined using the software GraphPad Prism 7—nonlinear regression (Y = Bottom + (Top − Bottom)/(1 + 10^((LogIC-X) × HillSlope)). The IC_25_ was determined using the online tool provided by AAT Bioquest. The data are presented as the averages of the repetitions with the standard error of the mean (SEM). The data were analyzed with a one-way analysis of variance (Tukey multiple comparisons of means, R Project version 4.0., Lucent Technologies, NJ, USA), where the differences were considered statistically significant when *p* < 0.05.

### 4.8. Ultrahigh-Pressure Liquid Chromatography-Mass Spectrometry (U-HPLC/MS) Analysis

The U-HPLC/MS analysis was performed by the Laboratory of Mass Spectrometry at the Central Laboratory of UCT. The measurements were performed on a Luna C18 Phenomenex 150 × 2 mm, 3.1 µm column on Thermo LC-MS LTQ—Orbitrap Velos spectrometer in both positive and negative modes. A gradient of 100% water (+0.1% formic acid) to 100% methanol (+0.1% formic acid) was used as the mobile phase over a time of 15 min. The data obtained were analyzed using Xcalibur 2.2. Molecular ion adducts such as [M + H]+, [M + Na]+, [2M + H]+, [2M + Na]+, [M − H]−, [2M − H]−, [2M − H + Na]− were manually identified in order to determine the molecular ion [M]. Once the major molecular ions were identified, a molecular formula (MF) was generated using Xcalibur, and a search using the [M] and MF filter was performed on Scifinder (https://scifinder-n.cas.org/) to identify potential compounds.

## 5. Conclusions

Our findings showed that arrayan and peumo fruits methanol extracts inhibit the growth of both *S. aureus* and *P. aeruginosa* pathogenic bacteria. Furthermore, the fruit extracts were effective in controlling the growth of the methicillin-resistant strain of *S. aureus*. The extraction method for obtaining a high content of antioxidant molecules in the extracts is an interesting opportunity for the treatment of tropical diseases. More studies, including the inherent stability of these extracts, are necessary before their future use in medicine.

## Figures and Tables

**Figure 1 antibiotics-09-00444-f001:**
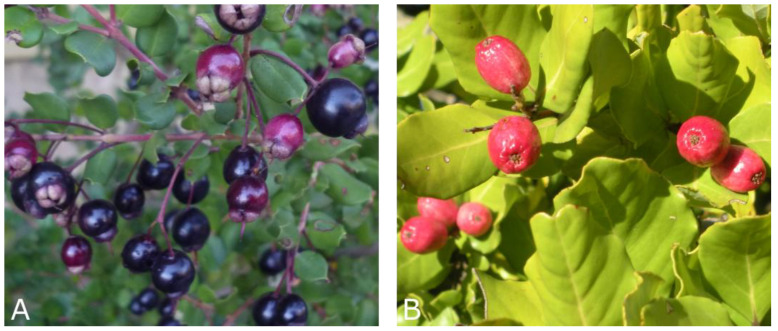
Branches with leaves and fruits of arrayan and peumo. (**A**) Arrayan [*Luma apiculata* (DC.) Burret.] *; (**B**) Peumo [*Cryptocarya alba* (Molina) Looser] **. Photography credit to Carlos R. Figueroa (*) and Lida Fuentes (**).

**Figure 2 antibiotics-09-00444-f002:**
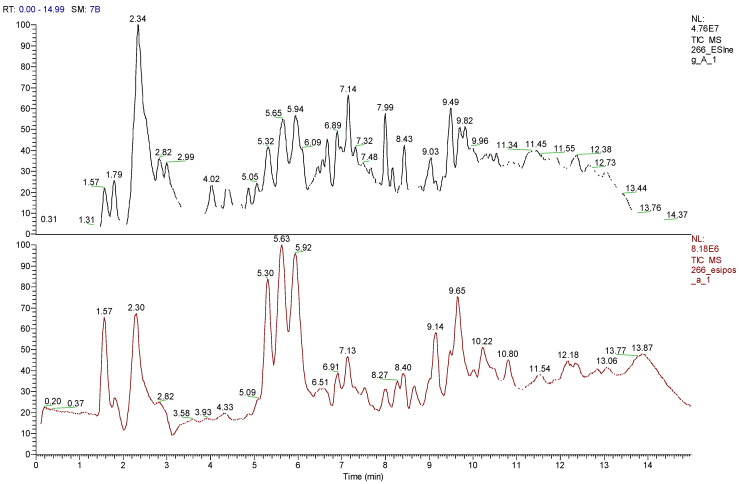
Total ion chromatogram (TIC) for arrayan. TIC of arrayan extract is shown in negative and positive modes.

**Figure 3 antibiotics-09-00444-f003:**
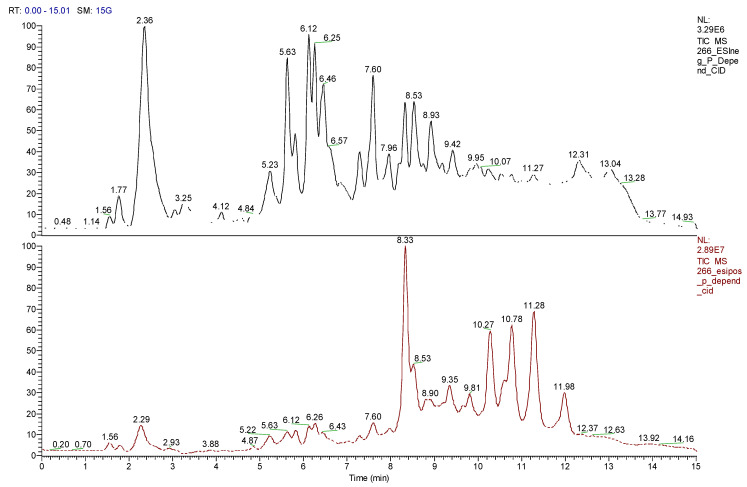
Positive and negative mode TIC spectrum for peumo. TIC of peumo extract is shown in negative and positive modes.

**Figure 4 antibiotics-09-00444-f004:**
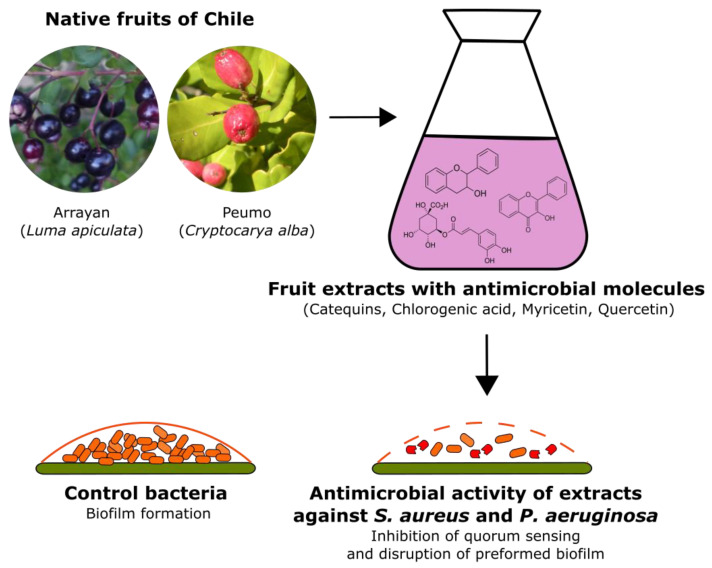
Graphical abstract of antimicrobial activity of fruit extracts of arrayan and peumo. Both extracts inhibit the growth of *S. aureus* and *P. aeruginosa* pathogenic bacteria potentially associate to its high content of antioxidant molecules. More details are presented in the text. Chemical structures credits [24].

**Table 1 antibiotics-09-00444-t001:** Inhibition of multidrug-resistant bacterial strains. Commercial antibiotics oxacillin or gentamicin served as the positive controls for *S. aureus* and *P. aeruginosa*, respectively.

Sample	IC_50_ (mg/mL)			
Extract/Compound	*S. aureus* (ATCC 25923)	*S. aureus* (NEM 449)	*P. aeruginosa* (ATCC 27853)	*P. aeruginosa* (NEM 986)
	sensitive strain	resistant strain *	sensitive strain	resistant strain **
Peumo	0.533 ± 0.018 ^c^	0.557 ± 0.034 ^c^	>1	0.778 ± 0.004 ^b^
Arrayan	0.354 ± 0.007 ^b^	0.385 ± 0.013 ^b^	>1	>1
Antibiotics	0.0046 ± 0.0002 ^a^	0.040 ± 0.002 ^a^	0.0002 ± 0.000	>0.150 ^a^

Data represent the average of four repetitions with corresponding standard errors of the mean. The data were analyzed with analysis of variance (ANOVA) (*p* < 0.05) and the statistical significances within one bacterial strain were denoted by different letters (^a,b,c^). * Resistant to gentamicin, clindamycin, erythromycin, chloramphenicol, vancomycin, ciprofloxacin, methicillin, penicillin G, cefotaxime, and tetracycline. ** Resistant to gentamycin, ciprofloxacin, tetracycline, chloramphenicol, penicillin G, and erythromycin.

**Table 2 antibiotics-09-00444-t002:** Modulation of the drug-resistant phenotype in bacteria by arrayan and peumo extract. The fold is expressed as a ratio of IC_50_ of antibiotics and IC_50_ of antibiotics with the addition of extract at IC_25_ concentration.

Sample	Fold	
Extract	*S. aureus* (NEM 449)	*P. aeruginosa* (NEM 986)
	resistant strain	resistant strain
Peumo	0.388 ± 0.0002	0.547 ± 0.0003
Arrayán	0.865 ± 0.0006	0.485 ± 0.0007

0.5 < FOLD > 1 indifferent effect, 0.5 > FOLD antagonistic effect, 1 < FOLD > 2 additive effect, 2 < FOLD synergistic effect.

**Table 3 antibiotics-09-00444-t003:** Inhibition of biofilm formation.

Sample	IC_50_ (mg/mL)	
Extract	*S. aureus* (ATCC 25923)	*P. aeruginosa* (ATCC 27853)
Peumo	0.473 ± 0.028 ^b^	0.346 ± 0.013 ^b^
Arrayan	0.229 ± 0.017 ^a^	0.288 ± 0.021 ^a^

Data represent the average of 16 repetitions with corresponding standard errors of the mean. The data were analyzed with ANOVA (*p* < 0.05) and the statistical significances within one bacterial strain were denoted by different letters (^a,b^).

**Table 4 antibiotics-09-00444-t004:** Disruption of mature biofilm.

Sample	IC_50_ (mg/mL)	
Extract	*S. aureus* (ATCC 25923)	*P. aeruginosa* (ATCC 27853)
Peumo	No activity	0.586 ± 0.042 ^a^
Arrayan	No activity	0.559 ± 0.040 ^a^

Data represent the average of 16 repetitions with corresponding standard errors of the mean. The data were analyzed with ANOVA (*p* < 0.05) and the statistical significances within one bacterial strain were denoted by different letters.

**Table 5 antibiotics-09-00444-t005:** Inhibition of quorum sensing.

Sample	IC_50_ (µg/mL)			
Extract	AI-1 strain BAA 1118 (G−)		AI-2 strain BAA 1119 (G+, G−)	
	viability	communication	viability	communication
Peumo	165.7 ± 21.8 ^a^	25.6 ± 0.3 ^a^	147.4 ± 2.3 ^a^	96.2 ± 7.4 ^b^
Arrayan	447.7 ± 71.6 ^b^	127.2 ± 5.9 ^b^	111.5 ± 15.3 ^a^	39.9 ± 3.9 ^a^

Data represent the average of three repetitions with corresponding standard errors of the mean. The data were analyzed with ANOVA (*p* < 0.05) and statistical significances within one bacterial strain were denoted by different letters (^a,b^).

**Table 6 antibiotics-09-00444-t006:** Identification of compounds from arrayan fruits by liquid chromatography coupled with mass spectrometry (LC-MS) and MS/MS data. The principal peaks were individually analyzed, and the potential compounds were analyzed. RT, retention time; MF, molecular formula.

RT (min)	[M+X]^+^ (m/z)	[M−X]^−^ (m/z)	[M] (m/z)	Fragments	MF	Tentative Compound
**4.02**	-	466.0311 [M-H]^−^	467	169.0149, 211.0259, 271.0476	C_26_H_10_O_9_	Unidentified
	-	933.0699 [2M-H]^−^				
**4.35**	-	466.0311 [M-H]^−^	467	156.1025, 184.0957	C_26_H_10_O_9_	Unidentified
	935.0790 [2M+H]^+^	933.0699 [2M-H]^−^				
**5.05**	307.0813 [M+H]^+^	305.0685 [M-H]^−^	306	125.0247, 137.0248, 151.0406, 167.0356, 179.0358, 219.0675, 221.0467	C_15_H_14_O_7_	Epigallocatechin Gallocatechin
	-	611.1443 [2M-H]^−^				
**5.29**	465.1029 [M+H]^+^	463.0915 [M-H]^−^	464	−301.0375, 337.0589	C_21_H_20_O_12_	Quercetin 3-glucosideMyricetin Hyperoside Isoquercetrin
**5.63**	479.1188 [M+H]^+^	477.1071 [M-H]^−^	478	−315.0533	C_22_H_22_O_12_	Isohamnetin-3-*O*-β-d-galactoside
	449.1083 [M+H]^+^	447.0964 [M-H]^−^	448		C_21_H_20_O_11_	Quercitrin Isoorientin Luteolin 7-*O*-glucoside
**5.92**	493.1346 [M+H]^+^	491.1230 [M-H]^−^	492	−169.0150, 305.0678, 331.0481	C_23_H_24_O_12_	Unidentified
**6.88**	481.0974 [M+H]^+^	479.0858 [M-H]^−^	480	+245.0453, 263.0559, 273.0403, 319.0449	C_21_H_20_O_13_	Myricetin 3-*O*-galactoside
**7.14**	465.1027 [M+H]^+^	463.0912 [M-H]^−^	464		C_21_H_20_O_12_	myricitin Quercetin 3-glucoside Quercetin 3-*O*-β-d-allopyranoside Isoquercitrin Hyperoside
**7.99**	599.2699 [M+H]^+^	597.2589 [M-H]^−^	598	−271.0476, 313.0585, 485.1695	C_29_H_42_O_13_	Unidentified
	621.2517 [M+Na]^+^	1195.5236 [2M-H]^−^				
**9.14**	387.1803 [M+H]^+^	-	386	289.1053	C_22_H_27_O_6_	Unidentified
	409.1621 [M+Na]^+^	-				
**9.45**	225.1121 [M+H]^+^	223.0987 [M-H]^−^	224	−179.1085 +139.0395, 155.0343, 207.1024	C_12_H_16_O_4_	Unidentified
**9.51**	503.3371 [M+H]^+^	501.3255 [M-H]^−^	502	+139.0402, 155.0352, 165.0561, 207.1035	C_30_H_46_O_6_	Guavenoic acid Guavalanostenoic acid
	525.3193 [M+Na]^+^	1003.6580 [2M-H]^−^				
**9.63**	415.2113 [M+H]^+^		414	+303.1211	C_24_H_30_O_6_	Unidentified
	437.1931 [M+Na]^+^					
	851.3975 [2M+Na]^+^					

**Table 7 antibiotics-09-00444-t007:** Identification of compounds from peumo fruits by liquid chromatography coupled with mass spectrometry (LC-MS) and MS/MS data. The principal peaks were individually analyzed, and the potential compounds were analyzed. RT, retention time; MF, molecular formula.

RT (min)	[M+X]^+^ (m/z)	[M−X]^−^ (m/z)	[M] (m/z)	Fragments	MF	Tentative Compound
**2.37**	-	341.1032 [M-H]^−^ 683.2142 [2M-H]^−^	342	−89.0227, 101.0226, 119.0330, 143.0329, 161.0433, 179.0537	C_19_H_18_O_6_	Unidentified
**5.23**	579.1509 [M+H]^+^	577.1362 [M-H]^−^ 599.1181 [M-Na]^−^	578	−289.0743, 407.0809, 425.0913, 451.1071	C_30_H_26_O_12_	Procyanidin B_1_ Procyanidin B_2_
**6.11**	355.1033 [M+H]^+^	353.0875 [M-H]^−^ 707.1821 [2M-H]^−^	354		C_16_H_18_O_9_	Chlorogenic acid 4-Caffeoylquinic acid
**6.24**	-	289.0718 [M-H]^−^ 579.1511 [2M-H]^−^	290	−179.0357, 205.0516. 245.0811	C_15_H_14_O_6_	Catechin, Epicatechin
**6.38**	470.1667 [M+H]^+^ 492.1486 [M+Na]^+^	468.1505 [M-H]^−^ 937.3083 [2M-H]^−^	469		C_21_H_27_O_11_N	Unidentified
**6.46**	355.1030 [M+H]^+^ 731.1821 [2M+Na]^+^	353.0876 [M-H]^−^ 707.1880 [2M-H]^−^	354	−191.0568	C_16_H_18_O_9_	Analogue of chlorogenic acid 4-Caffeoylquinic acid
**7.29**	465.1034 [M+H]^+^	463.0881 [M-H]^−^	464	301.0370	C_21_H_20_O_12_	Quercetin 3-glucoside Quercetin 3-*O*-β-d-allopyranoside Isoquercitrin Hyperoside
**7.58**	449.1084	447.0930	448	301.0371	C_21_H_20_O_11_	Quercetin 3-*O*-α-d-rhamnopyranoside Quercitrin Kaempferol 3-*O*-galactoside Quercetin 3-*O*-β-d-rhamnoside Astragalin Orientin
**7.94**	463.1241 [M+H]^+^	461.1087 [M-H]^−^ 923.2253 [2M-H]^−^	462		C_22_H_22_O_11_	Isorhamnetin-3-*O*-rhamnoside Luteolin 7-O-glucuronide
**8.34**	-	359.1504	360	313.1465, 327.1466, 341.1624	C_20_H_24_O_6_	(+)-Lariciresinol 4-*O*-Methylcedrusin
**8.52**	-	343.1766	344	165.0564, 255.1717, 297.1727	C_17_H_28_O_7_	Unidentified

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
