# Peer review of "Antimicrobial Activity of Extracts of Two Native Fruits of Chile: Arrayan (Luma apiculata) and Peumo (Cryptocarya alba)"

_antibiotics, 2020, doi:10.3390/antibiotics9080444_

Round 1
Reviewer 1 Report
In the manuscript “antibiotics-859364 the authors reported the antimicrobial activity of extracts of two Chilean trees: arrayan and Peumo. The authors studied the antimicrobial activity on sensitive or multidrug-resistant bacterial strains, their ability to inhibit or destroy biofilm as well as to inhibit quorum sensing.
The paper represents the scope of the journal and the scientific procedures have been well conducted. For this reason it may be accepted for publication.
Some minor modifications:
- Rows 349-356: the ref. 3 is missing. Please check carefully the list of references.
Author Response
Response to Reviewer 1
Q1: Rows 349-356: the ref. 3 is missing. Please check carefully the list of references.
A1: The reference was corrected.
We are very grateful to Reviewer 1 for the helpful comments.
Reviewer 2 Report
Overall, the paper is solid and worthy of publication. It is easy to read and understand, and only one major issue was found. In the methods, the “Inhibition of biofilm formation” and “Disruption of mature biofilm” assay are identical. My guess is the “Inhibition of biofilm formation” protocol is incorrect as it does not make sense to grow the bacteria for 24 h in the 96-well plate prior to adding test samples in this assay. This should be fixed or clarified before acceptance.
Minor issues found include the following:
Line 17 – “halving the biofilms formation mature biofilms consisting” is awkward to read
Line 20 – IC50 needs units
Table 5 – RLU may need defined
Line 216 – “none alkaloid was identified” is awkward to read, u-HPLC/MS should be U-HPLC/MS to be consistent
Line 236 – “Figure 1, A and D” should be A and B
Tables – formatting of tables is inconsistent (alignment of titles, borders in cells, etc.), reference to tables is text is inconsistent (sometimes it is BOLD while other places it is not)
Author Response
Response to Reviewer 2
Q1: In the methods, the “Inhibition of biofilm formation” and “Disruption of mature biofilm” assay are identical. My guess is the “Inhibition of biofilm formation” protocol is incorrect as it does not make sense to grow the bacteria for 24 h in the 96-well plate prior to adding test samples in this assay. This should be fixed or clarified before acceptance.
A1: Thanks to the reviewer for the warning, the method has been corrected.
Q2: Line 17 – “halving the biofilms formation mature biofilms consisting” is awkward to read
A2: Corrected.
Q3: Line 20 – IC50 needs units
A3: Solved.
Q4: Table 5 – RLU may need defined
A4: RLU was changed for communication. RLU was defined in methodology.
Q5: Line 216 – “none alkaloid was identified” is awkward to read, u-HPLC/MS should be U-HPLC/MS to be consistent
A5: Corrected.
Q6: Line 236 – “Figure 1, A and D” should be A and B.
A6: Corrected.
Q7: Tables – formatting of tables is inconsistent (alignment of titles, borders in cells, etc.), reference to tables is text is inconsistent (sometimes it is BOLD while other places it is not)
A7: Formatting of the tables was corrected.
We are very grateful to Reviewer 2 for the helpful comments
Reviewer 3 Report
Minor suggestions, but relevant for the readability of the manuscript, are reported below.
- Lines 81-84: Sentences reported the methods so could be move to Materials and Methods.
- Lines 93-100 (Table 1): In legend miss explanation for a, b, c apex. Please make the table more readable.
- Lines 98-100: please give information about concentrations in µg for each antibiotic.
- Line 110 (table 2): please follow the author’s instruction for title and legend.
- Lines 118 and 119: delete article “the” before the strains.
- Lines 121-124 (Table 3): in the legend materials and methods details were written, like under table 1, while explanation of apex a and b were done. Please make the table more readable.
- Lines 129-132 (Table 4): Please make the table more readable following the suggestions in lines 121-124
- Lines 146-149 (Table 5): Please make the table more readable following the suggestions in lines 121-124. In particular, please give explanation for apex a, b.
- Lines 260-262: please reported concentrations in µg for each antibiotic, if disk diffusion method (Kirby Bauer method) was performed. Please give informations about antibiotic susceptibility testing of strains and how was evaluated.
- References: please check the bibliography because in references number 2 are two bibliographic sources. The total number of references is 35 or 36 and in the text? Is the correspondence between text and references right?
Author Response
Response to Reviewer 3
Q1: Lines 81-84: Sentences reported the methods so could be move to Materials and Methods.
A1: Sentences were transferred to Methodology.
Q2: Lines 93-100 (Table 1): In legend miss explanation for a, b, c apex. Please make the table more readable.
A2: Table 1 was slightly changed, and footer was completed.
Q3: Lines 98-100: please give information about concentrations in µg for each antibiotic.
A3: The sentence has been modified to make more sense.
Q4: Line 110 (table 2): please follow the author’s instruction for title and legend.
A4: Solved.
Q5: Lines 118 and 119: delete article “the” before the strains.
A5: Solved.
Q6: Lines 121-124 (Table 3): in the legend materials and methods details were written, like under table 1, while explanation of apex a and b were done. Please make the table more readable.
A6: Solved.
Q7: Lines 129-132 (Table 4): Please make the table more readable following the suggestions in lines 121-124
A7: Solved.
Q8: Lines 146-149 (Table 5): Please make the table more readable following the suggestions in lines 121-124. In particular, please give explanation for apex a, b.
A8: Solved.
Q9: Lines 260-262: please reported concentrations in µg for each antibiotic, if disk diffusion method (Kirby Bauer method) was performed. Please give informations about antibiotic susceptibility testing of strains and how was evaluated.
A9: The drug-resistant/sensitive phenotype was determined by measuring of minimal inhibition concentration of each antibiotic and comparing the value with the clinical breakpoints defined by EUCAST. The methodology was corrected.
Q10: References: please check the bibliography because in references number 2 are two bibliographic sources. The total number of references is 35 or 36 and in the text? Is the correspondence between text and references right?
A10: The bibliography was corrected.
We are very grateful to Reviewer 3 for the helpful comments.